# On the Spatio-Temporal Dependence of Anomalies in the Atmospheric Electric Field Just around the Time of Earthquakes

**Yasuhide Hobara** [1,2,*]**, Mako Watanabe** [1]**, Risa Miyajima** [1]**, Hiroshi Kikuchi** [2]**, Takuo Tsuda** [1,2] **and Masashi Hayakawa** [3]

1. Graduate School of Informatics and Engineering, School of Informatics and Engineering, The University of Electro-Communications, 1-5-1 Chofugaoka, Chofu, Tokyo 1828585, Japan
2. Center for Space Science and Radio Engineering, The University of Electro-Communications, 1-5-1 Chofugaoka, Chofu, Tokyo 1828585, Japan
3. Hayakawa Institute of Seismo Electromagnetics Co., Ltd. (Hi-SEM), UEC (University of Electro-Communications) Alliance Center #521, 1-1-1 Kojima-cho, Chofu, Tokyo 1820026, Japan
* Correspondence: hobara@ee.uec.ac.jp

**Abstract:** In this study, we report atmospheric electric field (AEF) anomalies observed around the time of earthquakes (EQs) in Japan. Using a newly developed AEF observation network with three spatially separated stations in Japan (Chofu, Kakioka, and Iwaki), we conducted a study for two EQs that occurred within a few 100 km from the EQ epicenter under relatively good local weather conditions as shown by a local all-sky camera and weather information. Time series and wavelet analyses of the AEF indicate that fluctuation anomalies in the AEF with periods of 10–60 min and larger than 70 min were observed from a few hours before up to a few hours after the occurrence of the EQs. The lag in the onset time increased with increasing distance from the EQ epicenter to the field site. The above-mentioned characteristics of these AEF fluctuation anomalies were similar among the three stations, and therefore the observed AEF anomalies were considered to be an imminent precursor of EQs. The observed AEF anomalies were likely to be caused by internal gravity waves (IGWs) generated around the EQ epicenter a few hours before the EQ, passing over the field site while changing the AEF by changing the space charge density in the surface layer of the atmosphere.

**Keywords:** earthquake (EQ) precursors; atmospheric electric field (AEF); atmospheric gravity waves; lithosphere–atmosphere–ionosphere coupling (LAIC)

## 1. Introduction

During the last decade, enormous progress has been achieved in the field of seismo-electromagnetics (electromagnetic phenomena associated with earthquakes (EQs)), and observations have provided evidence that various electromagnetic phenomena appear not only in the lithosphere but also in the atmosphere and ionosphere (bottom part and upper F region) before an EQ as short-term EQ precursors [1–4]. Short term refers to the lead time of a few days to weeks. Many possible anomalies include electromagnetic radiation in a wide frequency range from DC/ULF to a higher frequency and different disturbances occuring in the atmosphere and ionosphere. In particular, the ionospheric perturbations not only in the bottom part, but also in the upper ionosphere, have attracted a lot of attention, and long-term analyses have found that they are statistically correlated with EQs and are considered to be promising candidates for short-term EQ prediction e.g., [5,6]. The most recent topic of seismo-electromagnetics is the elucidation of lithosphere–atmosphere–ionosphere coupling (LAIC); why and how the ionosphere is

perturbed due to the initial lithospheric disturbance before an EQ. Even in this situation, short-term EQ precursors are still a controversial subject because the association of those anomalies with EQs is not well evidenced. This association can be investigated with the help of criticality analysis [7,8].

In addition to the above-mentioned short-term EQ precursors, [1] has highlighted the importance of near-seismic effects or imminent precursors (this term will be used in subsequent sections) including seismic foreshock and ULF radiation. The word imminent refers to a lead time of less than one day, and therefore, it is much easier and more convincing to find a relationship between anomalies with EQs. A recent remarkable imminent precursor, the TEC (total electron contents) anomaly, was discovered immediately before the 2011 Tohoku EQ (~1 h before) by [9], and these anomalies have common characteristics in various locations around the world [10]. The quantitative relationship between spatial extent, amplitude, and lead time of anomaly and seismic magnitude obtained in this study is crucial information for future EQ prediction [10,11]. Hence, following this principle, we will focus on this imminent effect in the observation of the atmospheric electric field (AEF) with special reference to its temporal fluctuations.

Here we will describe why we pay attention to the observation of AEFs. Generally, the AEF anomaly with EQs has been studied less than other electromagnetic anomalies. Various papers by [12] and [13] reported the association of AEF anomalies with EQs. There have been papers published recently on the association of AEF anomalies (such as bay-like depressions) with EQs e.g., [14], but they are not all sufficiently convincing because they have not shown their spatial extent and dynamics due mainly to the single station observation. Therefore, we need sophisticated analysis methods such as those proposed in this paper, which will be of essential importance in identifying the seismo-related effects.

When discussing imminent precursors, coseismic effects must be mentioned. A typical example of this kind is coseismic ionospheric perturbations. These ionospheric disturbances have been studied for several decades since the pioneering work by [15] and TEC (total electron contents) observations detected a TEC anomaly starting 7 min after the 2011 Tohoku EQ with a magnitude of nine [16].

The construction of this paper is as follows: Section 2 consider the description of our newly developed AEF network observation. Section 3 will focus on the observational results on AEFs and wavelet results. Section 4 is the discussion and Section 5 is the summary and conclusions.

## 2. Observation and Analysis of AEFs

### 2.1. AEF Observation Network

In this paper, the vertical AEF data were obtained from our AEF observation network newly developed by the University of Electro-Communications (UEC) group consisting of three sensor locations, Chofu (CHF), Kakioka (KAK), and Iwaki (IWK). As shown in Figure 1, three field sites were located on the mainland (near Tokyo, Kanto region). Three sites were aligned in the northeast direction and the distance between each site was around 100 km. An electric field mill (BOLTEK-100 system) was used to record the vertical electric field continuously. We used an inverted downward-facing configuration to avoid precipitation noise and possible damage to the sensor unit. At all stations, the electric field waveforms were observed with a time interval of 0.5 sec (2 Hz), but the average value per minute was used for the data analysis. They were installed on the roofs of the buildings at CHF and IWK (height ~20 m) and were installed on the ground at KAK.

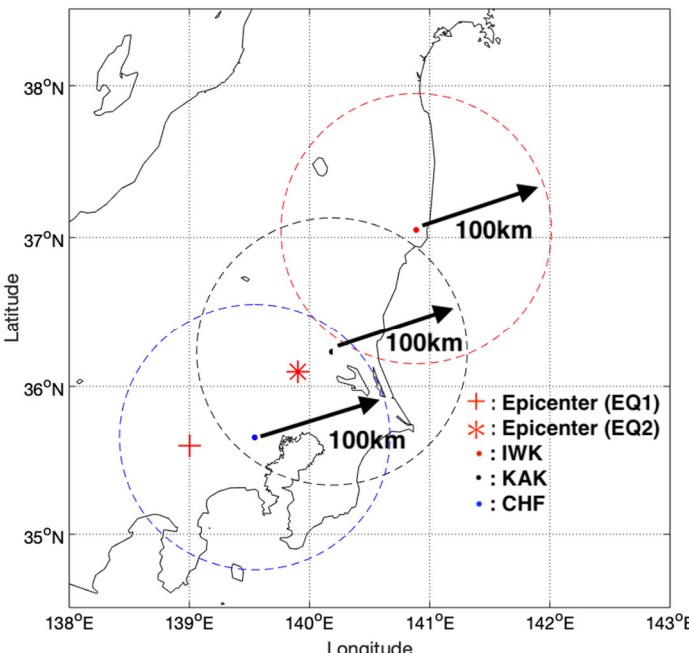

**Figure 1.** Relative locations of the three field sites (IWK, KAK, and CHF) for atmospheric electricity observation (shown as different coloured dots) and the epicenters of the two EQs (EQ1 is shown as a red plus and EQ2 is shown as a red diamond) referred to in this study. A circle with a radius of 100 km from each AEF site is shown as a dashed line.

*2.2. EQ and Meteorological Data*

EQ data were obtained from the Japan Meteorological Agency (JMA) website (http://www.jma.go.jp/jma/ accessed on 25 September 2022). We analyzed two recent EQs with moderate magnitudes and shallow depths. Table 1 summarizes the parameters of those EQs. EQ1 occurred in the early morning local time (06:37 JST on December 3) or at 21:37 UT on December 2, 2021, while EQ2 occurred around noon local time (12:31 JST on Deecmber 12) or at 03:31 UT on December 12, 2021. The relative locations of the two EQs were different as shown in Figure 1. The epicentral distances to the AEF sites are shown in Table 1. The epicenter of EQ1 was located on the west side of CHF and the further east the site was, the larger the distance from the epicenter . The epicenter of EQ2 was located between CHF and KAK (KAK was slightly closer to the epicenter than CHF) and the furthest site was IWK, which was located east of the epicenter. Both EQs are recognized as reverse fault types.

**Table 1.** Parameters of two EQs selected for this study.

| EQ No | Date | Time (UT) | Lat. [deg] | Long. [deg] | Mag. | Depth [km] | EQ Type | Distance (epicentric) CHF/KAK/IWK [km] | Weather |
|---|---|---|---|---|---|---|---|---|---|
| 1 | 20211202 | 2137 | 35.6 | 139.0 | 4.8 | 19 | R | 53/129/235 | fine |
| 2 | 20211212 | 0331 | 36.1 | 139.9 | 5.0 | 50 | R | 77/58/146 | fine |

AEFs were considered severely disturbed by local convective activities such as thunderstorms and clouds over AEF sites, so we used the following meteorological information to examine whether we were at relatively quiet meteorological conditions around the time of the two EQs [17]. The all-sky camera was installed next to the AEF sensor on the roof of the building at CHF. Images were taken every minute throughout the day and night, and local weather conditions (i.e., clear skies, cloud coverage, rain, over CHF) was obtained. The spatial distributions of clouds were obtained from the

geostationary meteorological satellite (MT-SAT). Both visible and infrared images were used to identify convective activities around Japan, including EQ epicenters and all AEF sites obtained from the JMA website [https://www.jma.go.jp/jma/menu/menureport.html accessed on 25 September 2022]. Moreover, weather maps and wind information were obtained on the day of the EQs (also from the JMA website) to identify the general weather conditions.

## 3. Observational Results of AEFs

### 3.1. AEF Variations during EQ1

Figure 2 demonstrates the diurnal variations of the AEF from November 25 (8 days before EQ1) to December 6 (3 days after EQ1) observed at our AEF stations, (a) CHF, (b) KAK, and (c) IWK. The days in gray curves include the periods with significant disturbances due to local bad weather. The time of EQ1 (06:37 JST), which occurred in the early morning local time, is shown as a vertical dashed line. The AEF variations for CHF and KAK on the day of EQ1 were not significant and are similar to other days except for heavily disturbed days, while the variation in IWK shows a few large drops at around 01:30, 02:30, and 06:30 JST. A very large bipolar change was seen at around 10 JST.

Figure 3 shows the time series (upper panel) and dynamic periodograms (lower panel) (the time interval up to 100 min) of the AEF 6 h before and 6 h after EQ1 for each AEF site ((a) CHF, (b) KAK, and (c) IWK). The time of the occurrence of EQ1 is shown as a vertical dashed curve. Wavelet analysis using Morlet wavelet was applied to each time-series AEF dataset (top panel) and the ordinate of the lower panel indicates the fluctuation period in a range from 20 to 100 min in the frequency range of IGW (internal gravity wave).

It was preferable for us to first consider the data at CHF (Figure 3a) because the epicenter of EQ1 was closest to the observing station of CHF and we can assume that the original agent leading to the generation of such fluctuations was likely to be around the EQ epicenter. Figure 3a suggests a global structure of three red bands (higher intensity fluctuations). The first red band with a period greater than 80 min was from 6 h before (even earlier than this time based on the one-day data of Figure 4) to 3 h after the EQ. The second clear enhancement of the wave energy with a time period of 40–50 min appears just around the EQ (two hours before and ~1 h after the EQ) in the wavelet periodogram corresponding to the wave-like oscillations in the time series. This second wavelet accompanies a minor wavelet at around 7:10 JST in the CHF time series, corresponding to the enhancement at around 20 mins in the wavelet periodogram. These structures seem to suggest the presence of a precursory nature.

Similar variations to CHF are seen both in the time series and the periodogram for KAK. In KAK wave-like oscillations starting around EQ1 began from 6 JST (1 h later than CHF) and continued up to 10 JST, which may have corresponded to the wave energy with a much longer time period than in CHF, 70–90 min in the wavelet periodogram. While the single positive enhancement was seen at 8:10 JST in the time series corresponding to the enhancement of the wave energy in 25 min.

The time series in IWK exhibited a series of sharp drops and bipolar changes corresponding to a known reason other than seismic activity, i.e., local weather disturbances (mentioned in the next subsection), which may seriously contaminate the wavelet spectrogram and so a simple linear interpolation has been made for the time period of the drop at 6:30 and the bipolar pulse around 10 JST. Except for the above-mentioned variations, wave-like oscillations started around EQ1 starting from 8 h JST (2 h later than CHF) and continued up to 13 JST in two time periods (around 40 and 70–90 min), which corresponded to the wave energy with much longer time periods than in CHF but with a similar period as the KAK 70–90 min in the wavelet periodogram.

The same type of variations as Figure 3 but with a longer time on the day of EQ1 (24 h) are shown in Figure 4. Distinctive wave-like oscillations around EQ1 (red vertical

dashed line) are still identifiable in all three stations (especially CHF and KAK, and are less distinctive because of the many pulses due to local weather disturbance in IWK).

One of the most important points here is the difference in onset timing between the three stations. Onsets of these oscillations become later in time with a larger epicenter distance to the station (Table 1). Moreover, the duration of the oscillation is close to 3–4 h among the stations.

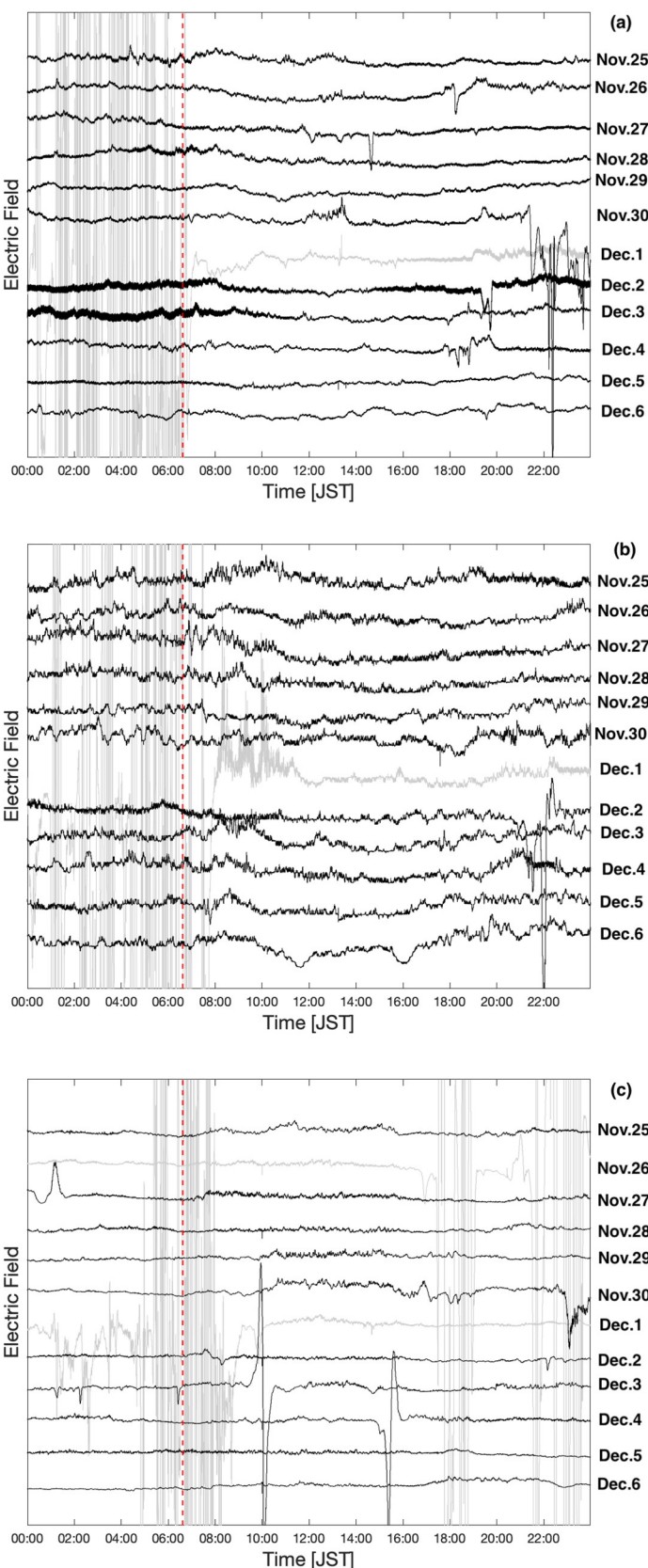

**Figure 2.** Daily time series of the AEF from 8 days before up to 3 days after EQ1 on December 3: (**a**) Chofu (CHF), (**b**) Kakioka (KAK), and (**c**) Iwaki (IWK). Heavily disturbed days due to bad local weather conditions are shown as gray curves.

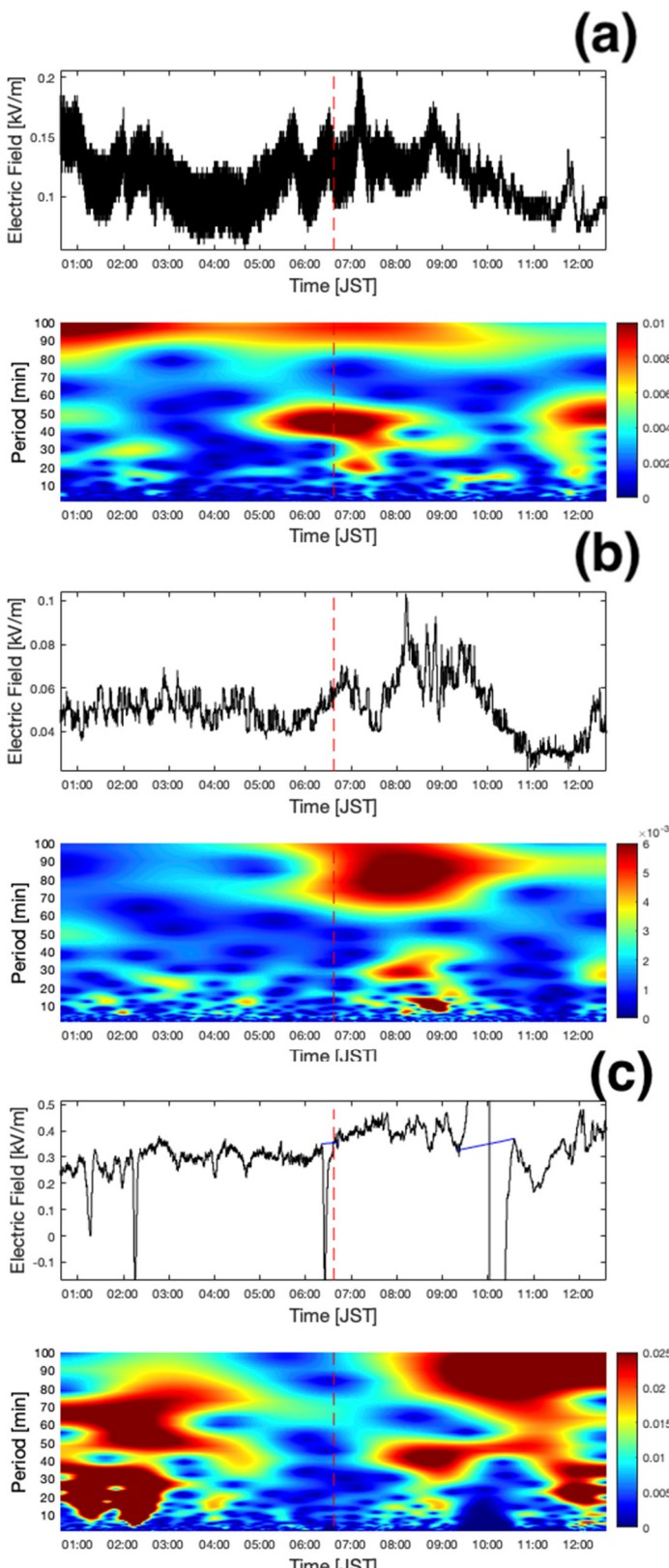

**Figure 3.** Time series and dynamic periodograms of the AEF with the period from up to 100 min in the interval from 6 h before to 6h after EQ1: (**a**) Chofu (CHF), (**b**) Kakioka (KAK), and (**c**) Iwaki (IWK).

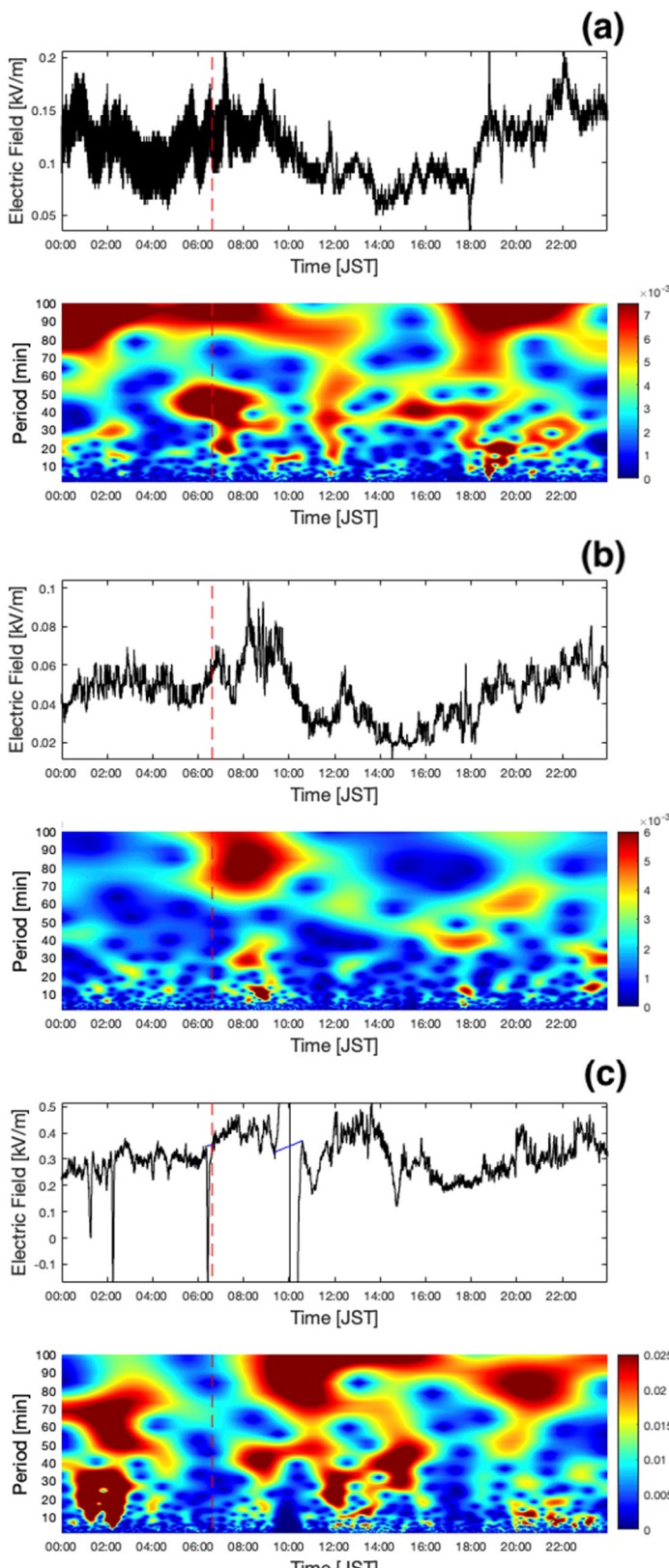

**Figure 4.** Time series and dynamic periodograms of the AEF for the period interval up to 100 min on the day of EQ1 (24 h): (**a**) Chofu (CHF), (**b**) Kakioka (KAK), and (**c**) Iwaki (IWK).

### 3.2. Local Weather and Space Weather Conditions during EQ1

AEFs are known to vary with local convective activity such as overlaying clouds, rain, wind, and space weather effects. In order to find any significant seismogenic signatures, we needed to exclude the meteorological and space weather effects. Accordingly, we examined the local meteorological and space weather conditions during the EQ time.

Figure 5 shows the all-sky camera images at CHF around the time of EQ1 ((a) 1.5 h before EQ1 (5 h JST), (b) 30 min before EQ1 (6 h JST), and (c) 30 min after EQ1 (7 h JST)). These figures show that the sky was clear around the time of EQ1 and the weather conditions were confirmed to be fair.

In Figure 6a,b, the weather map (9 h JST) and IR (infrared) image (6 h JST) from the meteorological satellite are shown from the same day as EQ1. There was no thick cloud cover around the epicenter and the surrounding observation stations. Moreover, a weather report from the nearest weather station from each observation site was obtained (Tokyo for CHF, Tsuchiura for KAK, and Onahama for IWK). With the exception of Onahama station, the weather was fine with a moderate wind speed (<2 m/s) (fair weather condition according to [17]) and so little effect on AE would be expected from the local weather conditions. In Onahama weather varied for a short time, it was rainy and cloudy in the early morning for 1–2 h, fine for 3–6 h, cloudy for 7 h, then fine for 8 h, and cloudy with strong wind for 9–12 h. Several large drops were seen in IWK, which corresponded to the bad weather in Onahama.

Around the date of EQ1, space weather conditions were rather quiet and so little effects would be expected for the AEF from space because Dst ≤ 40 nT, Kp ≤ 4, and no major solar events (e.g., SPEs (solar proton events) and solar flares) were observed.

(a) (b) (c)

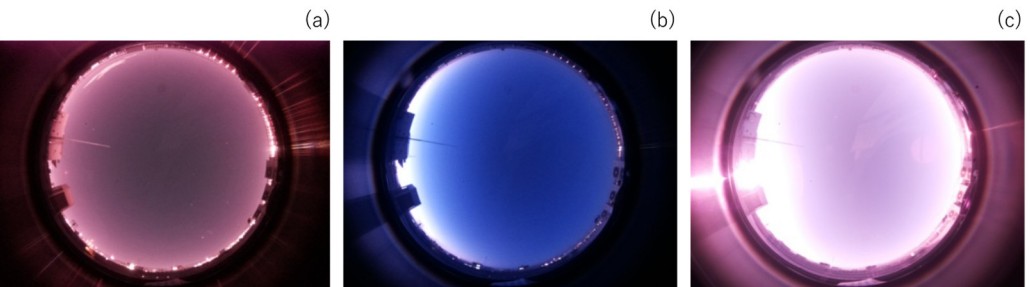

**Figure 5.** All-sky camera images at CHF around the time of EQ1: (**a**) 1.5 h before the EQ, (**b**) 30 min before the EQ, and (**c**) 30 min after the EQ.

(a) (b)

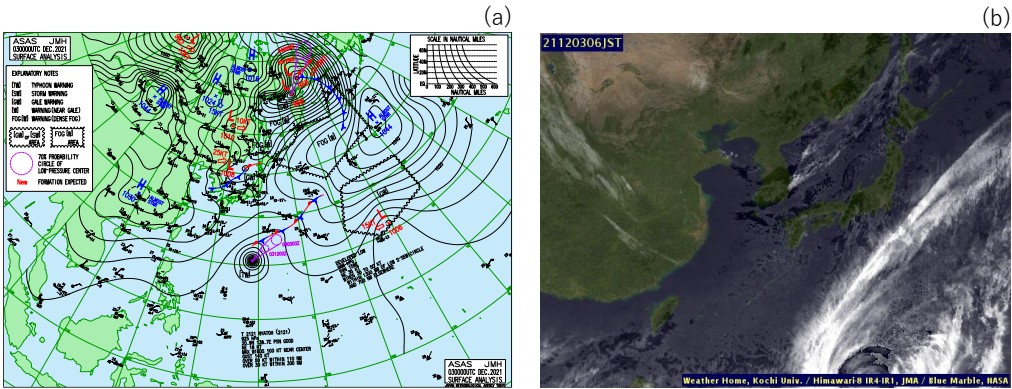

**Figure 6.** (**a**) A weather map and (**b**) a IR weather satellite image around the time of EQ1.

### 3.3. AEF Variations during EQ2.

In Figure 7, we present diurnal variations of the AEF from December 6 (6 days before EQ2) to December 18 (6 days after the EQ2) observed at our AEF stations, (a) CHF, (b)

KAK, and (c) IWK. Similar to Figure 2, the days in gray curves include the days with heavy disturbances due to local bad weather. The time of EQ2 (12:31 JST), which occurred shortly after noon, is shown as a vertical dashed line. The AEF variations for all three AEF stations do not show heavy disturbances on the day of EQ1 (December 12).

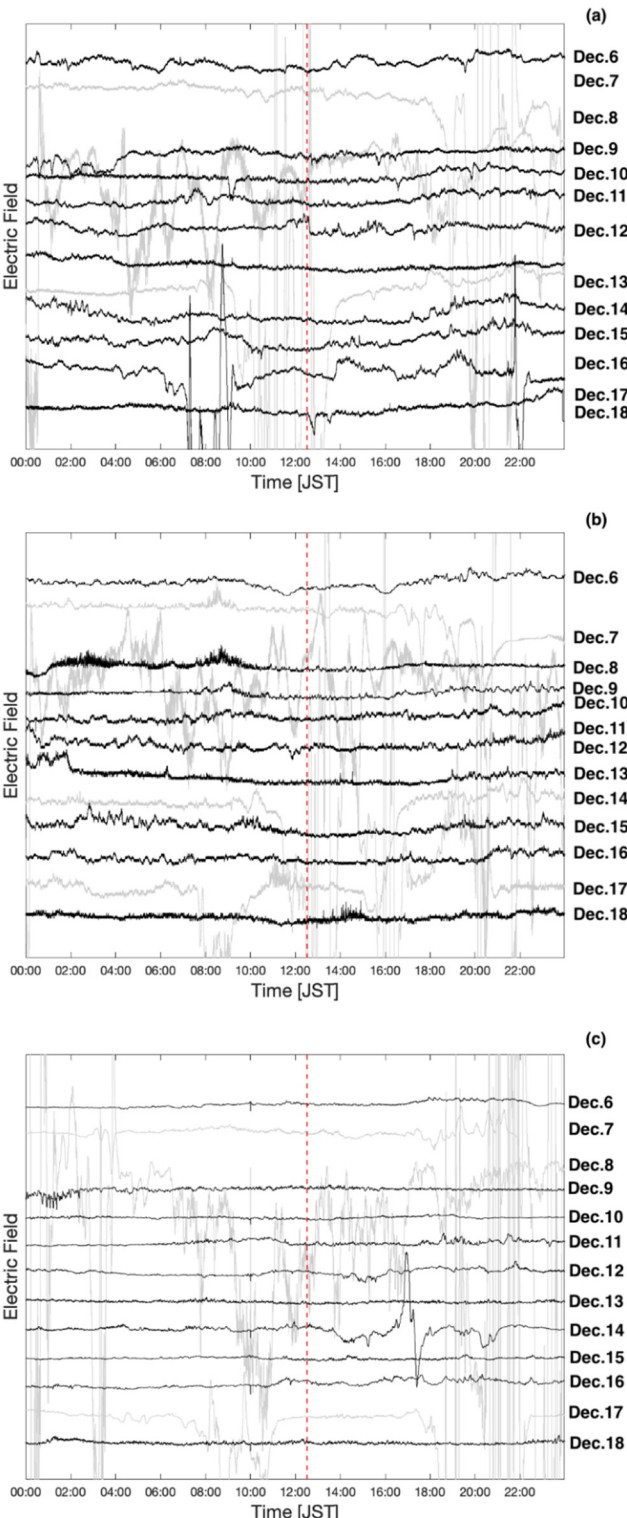

**Figure 7.** Daily time series of the AEF from 6 days before to 6 days after EQ2 on December 12: (**a**) Chofu (CHF), (**b**) Kakioka (KAK), and (**c**) Iwaki (IWK). Heavily disturbed days due to bad local weather conditions are shown as gray curves.

Figure 8 shows the time series (upper panel) and dynamic periodograms (lower panel) (time interval up to 100 min) of the AEF 6 h before and 6 h after EQ2 for each AEF site ((a) CHF, (b) KAK, and (c) IWK). The time of the occurrence of EQ2 is shown as a vertical dashed line.

As before, we considered first the results at the observing station of Kakioka (Figure 8b), because the information at this station was likely to provide us with the most invaluable information of the original cause of those fluctuations. We identified quite a similar three-red-band structure in KAK as the previous case, shown in Figure 8b. The longest period wavelet with a period of greater than 80 min started to appear at around 6 h JST (about 5 h before the EQ) and continued until one hour after the EQ. Then, the second wavelet with a period of 30–40 min appeared about 2 h before the EQ and remained even after the EQ, with a minor wavelet of a shorter period (30–40 min) than that of the second wavelet. These fluctuations are highly likely to be an imminent precursor to this EQ.

Remarkable enhancement of the AEF amplitude with an oscillation in time series in CHF was identified from 11 h (1.5 h before the occurrence time of EQ2) and continued for 1.5 h up to EQ2. After EQ2, amplitude dropped but periodical enhancement continued further. Wave energy with a time period of 40–50 min in the wavelet periodogram corresponded to the enhancement of the AEF with oscillations in the time series starting at around 11 h, and these wave-like oscillations continuing up to 14 h (up to 1.5 h after EQ2). Moreover, wave-like oscillations with a much longer time period beyond 100 min were identified from the spectrogram. These longer periodic oscillations started at the same time as those with a time period of 40–50 min but continued much longer up to 18 h.

Clear enhancements of wave energy with a time period of 30–40 min, similar to CHF, were also identified in KAK and IWK (Figure 8). The enhancement in KAK started 20 min earlier than the enhancement in CHF and lasted for 2 h until 12:40, while the enhancement in IWK started much later at around 13:30, and lasted for 3 h until 16:30. Furthermore, the wave energy beyond a 100-min time period was also identified both in KAK and IWK. They were mostly associated with the enhancement in a 40-min period.

As demonstrated in Figure 9, the above-mentioned enhancements were very isolated around the EQ and represent similar temporal dependence (time period and duration) on the day of EQ2. Similar to the EQ1, observed oscillations started later in time with larger epicenter distances to three stations (Table 1).

### 3.4. Local Weather and Space Weather Conditions during EQ2

We examined the meteorological conditions at every observation site around the time of EQ2. Figure 10 shows the all-sky camera images at CHF around the time of EQ2 ((a) 1.5 h before EQ2 (11 h JST), (b) 30 min before EQ2 (12 h JST), and (c) 30 min after EQ2 (13 h JST)). As shown in the figures, the sky was almost clear, and there were no thick clouds near the zenith around the time of EQ2.

In Figure 11a,b, the weather map (9 h JST) and IR image (12 h JST) from the meteorological satellite are shown on the day of EQ2. Although the low-pressure system was identifiable in the northern part of Japan, Kanto's southern part (including near Tokyo) seemed to be covered by a high-pressure system. The IR image indicated that there were no major clouds over the epicenter and surrounding observation stations CHF and KAK, while patchy clouds were seen over IWK around the time of EQ2. The weather report from the nearest weather station from each observation site (Tokyo for CHF, Tsuchiura for KAK, and Onahama for IWK) indicate that the weather was fine in CHF and KAK with a moderate wind speed (<2.5 m/s) and therefore there would be very little effect expected on the AEFs from the local weather conditions. In Onahama, the weather was fine at the time of EQ2 with no strong wind (~3 m/s). It was cloudy (13:30–16:00 h) with a wind speed of 3–5 m/s and therefore no major effect on the AEFs could be expected.

Around the date of EQ2, space weather conditions were rather quiet and so little effect was expected for the AEFs from space because Dst ≤ 10 nT, Kp ≤ 1, and no major high-energy solar event was observed.

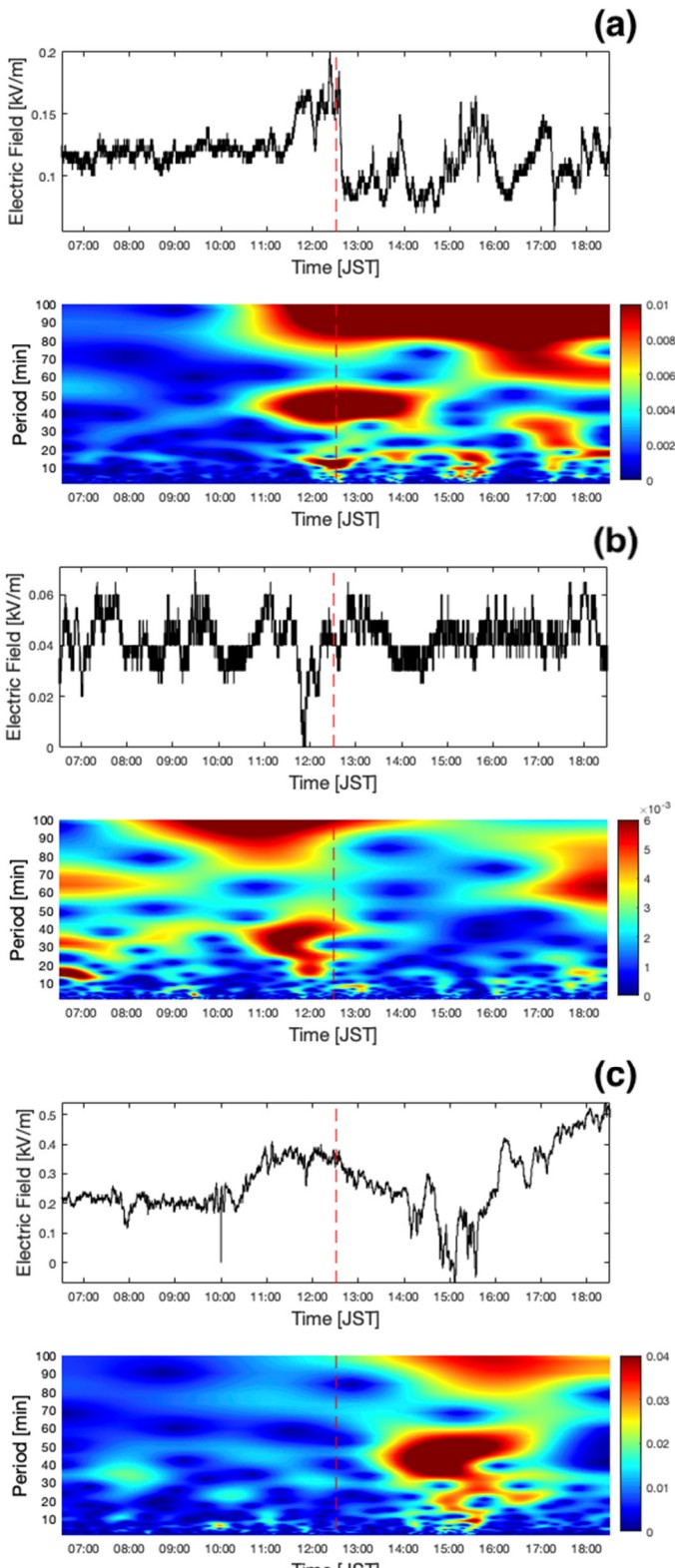

**Figure 8.** Time series and dynamic periodograms of the AEF with the period up to 100 min in the interval from 6 h before to 6h after EQ2: (**a**) Chofu (CHF), (**b**) Kakioka (KAK), and (**c**) Iwaki (IWK).

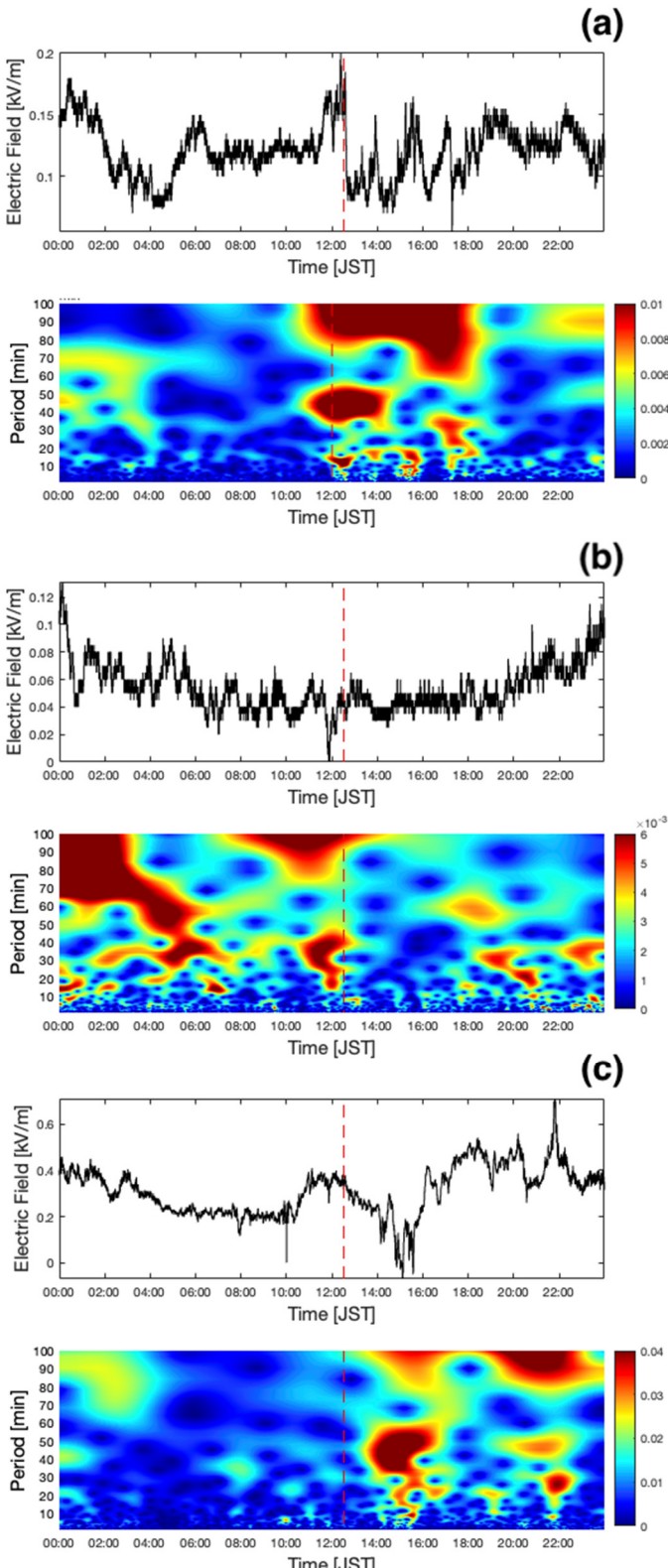

**Figure 9.** Time series and dynamic periodograms of the AEF for the period interval of up to 100 min on the day of EQ2 (24 h): (**a**) Chofu (CHF), (**b**) Kakioka (KAK), and (**c**) Iwaki (IWK).

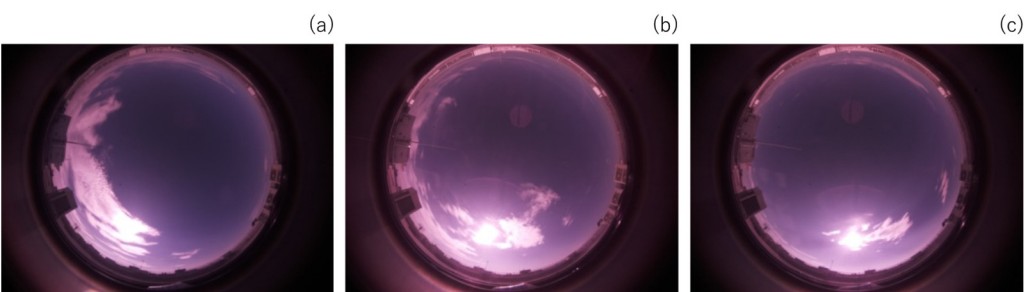

**Figure 10.** All-sky camera images at CHF around the time of EQ2: (**a**) 1.5 h before the EQ, (**b**) 30 min before the EQ, and (**c**) 30 min after the EQ.

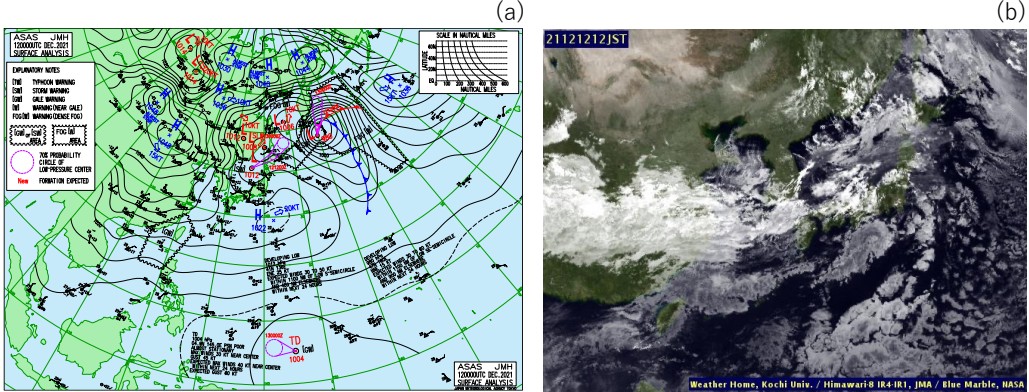

**Figure 11.** (**a**) Weather map and (**b**) IR weather satellite image around the time of EQ2.

## 4. Discussion

One of the most influential parameters affecting the AEF variations is the meteorological convective activities, therefore, we made considerable effort to exclude the local meteorological conditions at every AEF station on the days of the EQs from the data from both the ground and the satellite measurements. As a result, the observed large variabilities (typically sharp increases or decreases in AEF amplitude in time series) were found to be correlated well with a time of bad weather near the AEF observation station. These anomalies are probably due to the passage of the electric charges in the overlaying cloud, wind, and precipitations around the AEF sites. On the other hand, AEF anomalies around two seismic events were observed even during fair weather conditions (the sky was clear) which was proven by the all-sly camera in CHF with a light wind from the weather report. Moreover, little space weather effect was expected around the days of the two EQs. Therefore, we consider that the observed AEF anomalies in this paper are related to the EQs.

Most AEF anomalies related to EQs have focused on a short-term time window (a month before the occurrence of the EQs) e.g., [12,13,18,19], but in this case, the causality between the observed AEF anomaly and EQs is still relatively difficult to prove. So, this work has dealt with the AEF anomaly immediately before and after the EQs (i.e., imminent precursor). Imminent (6–7 h before the EQ) AEF anomalies have been rarely reported e.g., [20,21], and the observed anomalies are much easier to correlate with EQs. Moreover, for the first time, we successfully obtained the wavelet dynamic spectra and spatio-temporal dependence of AEF anomalies related to the two different EQs, shedding light on its generation mechanism.

Clear AEF anomalies were observed for two EQs that occurred at different locations within our AEF observation network consisting of three AEF sensors. The observed anomalies had remarkable similarities in temporal signatures between the two EQs. They were observed within a few hours around those EQs at different AEF stations with a recognizable time delay between stations for the same EQ. Although the anomalies were

recognized both in time series and corresponding wavelet spectrograms of AEFs, they were more pronounced in the frequency domain as an enhancement of energy in particular frequency (periodicity) ranges (as a wave-like oscillation) rather than the systematic increase or decrease of AEF amplitude. Two distinctive frequency bands were found, which were enhancements of energy with periodicity from 20–60 min and greater than 70 min. Moreover, the anomalies started later in time with larger epicenter distances to three stations for both EQs.

Here the mechanism of the LAIC process either for short-term or imminent EQ precursors has been studied extensively for the last ten years, and a few hypotheses have already been proposed e.g., [1,4]. As is given in [22], the first is a so-called chemical channel, in which emanation of radioactive radon and gases is considered to be the main player, leading to the modification of atmospheric conductivity and generation of the electric field, thereby driving a variation in the ionospheric plasma density (e.g., [23–25]). The second is an acoustic channel, in which atmospheric oscillations, including atmospheric gravity waves (AGWs) and acoustic waves, are excited by the precursory deformation of ground motion and gas emanation [5,26], propagating upwards to the lower and upper ionosphere and leading to a perturbation in the ionosphere [27–29]. The third is the electromagnetic channel [22], where electromagnetic waves generated in any frequency range propagate upwards into the ionosphere and magnetosphere, inducing particle precipitation into the upper atmosphere due to wave–particle interactions in the magnetosphere. Finally, a fourth electrostatic channel is proposed based on laboratory experiments, in which positive holes are generated when the ground of interest is stressed by the accumulated pressure [30]. These processes have been discussed extensively by different authors, but none of the above hypotheses have been evidenced by any observational data, requiring further studies until the process of LAIC is well understood. Many papers using subionospheric sounding with the VLF/LF technique have been published, providing a lot of indirect evidence for the AGW hypothesis [31–33].

Recently, there considerable attention has been given to the AGW channel. Chen et al. [34] reported that ground vibrations, atmospheric pressure, and total electron content varied from ~$10^{-3}$ to ~$10^{-2}$ Hz before the Luxian EQ with a magnitude of 6. They consider that the AGW was triggered by the observed ground vibrations. Additionally, recent studies by [27–29] have shown the presence of AGW activity in the stratosphere to be well correlated with VLF/LF ionospheric perturbations for the 2011 Tohoku EQ and the 2016 Kumamoto EQ. Furthermore, [1] has discussed the possibility of excitation of AGWs with the release of radon and different gases from the lithosphere.

In most previous works, a generation mechanism of AEF anomalies has been suggested, such as radon concentration changing the conductivity near the ground atmosphere (chemical and conductivity channels), which directly changes the AEF over the region where the anomalies occur [21]. However, our observed AEF anomalies seem to support the AGW hypothesis which can be linked to the stratospheric AGW activity for the following two reasons. The first reason is that the observed anomalies have a systematic time delay between the three AEF stations in start time and duration and similar time-frequency dependences. The larger the epicenter distance, the later the onset time of the anomaly. This observed fact implies the anomaly is generated around the epicenter and propagates outward. The second reason is that the observed anomalies are more similar to oscillations with a period significantly shorter than the local Brunt–Vaisala frequency (i.e., periodicity ~5 min) and then fall in the range of IGWs. Whatever the mechanism is, our paper has presented rather strong evidence of the excitation of IGWs from near-surface seismogenic disturbances.

## 5. Summary and Conclusions

In this study, we report AEF anomalies observed around the time of EQ occurrence in Japan. Using our newly developed AEF observation network consisting of three spatially separated field sites near the Kanto region in Japan, we examined the time series

of the AEF around the timing of two EQs with M~5 that occurred under fair weather conditions with different geolocations in the AEF network.

From the results of the data analysis, we found common features of atmospheric electric field anomalies for the two EQs: (1) AEF anomalies around the EQ epicenters start a few hours before the EQ and continue up to a few hours after the EQ. This behavior suggests the presence of EQ imminent precursors. (2) The AEF anomaly has two particular major fluctuation periods (10–60 min and greater than 70 min). (3) Quite similar general patterns of fluctuation spectra were detected also at the other two stations further away from the closest station, but the onset timing of those anomalies becomes later as the distance to the field site from the EQ epicenter increases.

The above-mentioned common features indicate that the observed AEF anomalies are considered to be an imminent precursor of an EQ. Since the oscillation period of the observed AEF anomalies falls in the period of internal gravity waves (IGWs) and the arrival timing is later at field sites with larger distances from the epicenter, the observed AEF anomalies are probably caused by IGW waves generated around the epicenter for up to several hours before the EQ, passing over the field site while causing the variability in the AEF by changing the space charge density in the atmosphere.

Our immediate future task is to develop a quantitative model of the relationship between observed IGWs, e.g., [35], and AFE anomalies to understand their physical mechanism as well as the generation mechanism of seismogenic IGWs. Then, we will conduct a long-term analysis investigating more EQ events to obtain the statistical properties of the relationship between the observed AEF and the EQ, paying particular attention to the time of their occurrence and the EQ locations. Additionally, we will investigate the corresponding lower ionospheric perturbation by subionospheric VLF/LF data to enable a comparison with the AEF results in this paper.

**Author Contributions:** Conceptualization, writing, methodology, data collection and curation, and formal analysis, Y.H.; data analysis and visualization, M.W.; software, R.M.; data collection and discussion, H.K. and T.T.; discussion, M.H. All authors have read and agreed to the published version of the manuscript.

**Funding:** This research received no external funding.

**Institutional Review Board Statement:** Not applicable.

**Informed Consent Statement:** Not applicable.

**Data Availability Statement:** Atmospheric electric field data and all-sky camera presented in this study are available on request from the corresponding author. Earthquake, MT-Sat, weather map, and ground weather data are available from https://www.jma.go.jp/jma/menu/menureport.html (accessed on 25 September 2022); Kp and Dst index data are available from the World Data Center for Geomagnetism, Kyoto (https://wdc.kugi.kyoto-u.ac.jp/wdc/Sec3.html (accessed on 25 September 2022)). Solar events were obtained from https://cdaw.gsfc.nasa.gov/CME_list/ (accessed on 25 September 2022).

**Acknowledgments:** The authors would like to acknowledge the JMA for the earthquake catalog, MT-Sat, and weather data. YH would like to thank Mr. Aoki of UEC for AEF installation in CHF, Tamura of Joint Co., Ltd. and Joko Service for assisting with the AEF measurement in Iwaki, Fukushima, and also thank Nagamachi, Kaito, Yamagiwa and Hirahara of Kakioka magnetic observatory for assisting with AEF observations in Kakioka, Ibaraki.

**Conflicts of Interest:** The authors declare no conflict of interest.

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
