# Peer review of "On the Spatio-Temporal Dependence of Anomalies in the Atmospheric Electric Field Just around the Time of Earthquakes"

_atmosphere, doi:10.3390/atmos13101619_

Round 1

Reviewer 1 Report

Measuring the variations of the atmospheric electric field (AEF) - the "fair weather field" is a methodologically complex task. The authors implemented AEF observations in three points of the seismic area. Much attention is paid to weather monitoring: all-sky cameras, data from geostationary satellites and the weather service. The results of observation of two medium-magnitude earthquakes are given.The obtained results give reason to cautiously state that AEF changes can be considered as short-term earthquake precursors: «imminent precursors». The frequencies of the variations fall into the range of frequencies of internal gravitational waves (IGW). The conclusion is interesting, although it is based on observations of only two earthquakes. The authors provide a plan for further research and ideas for modeling the observed processes and searching for possible physical mechanisms that can cause variations in AEF - radon emanation, piezoelectric effect of earth rocks, possibly others. The authority of the authors of the work led by S.Hayakawa in the field of seismic-electrical process research is very high, and this work is a continuation of their work of the past decades.  

Author Response

Thank you so much for your kind comments. We are going to increase the number of events to be analyzed and further investigate physical mechanisms.

Reviewer 2 Report

This review is about the article “On the spatio-temporal dependence of anomalies in the atmospheric electric field just around the time of earthquakes”; by Yasuhide Hobara et al.

In general, I found the manuscript very informative, clear, well written, and easy to follow. It should be considered for publication. However, the following minor points may be incorporated before publishing.

Figure 1 caption should be moved to the respective figure location.

Most of the figure labels are not clear. Better to enhance the readability and resolution of figures.

Author Response

Thank you so much for your kind comments and suggestions regarding the figures.

We have revised the manuscript in response to your comments. The position of the figure captions has been corrected and the figure resolution has been improved.

Reviewer 3 Report

This study includes research on a scientifically necessary and interesting subject. The research method and the results obtained can offer a glimmer of hope for the prediction of earthquakes. Answering the following questions about the study will contribute to the development of the study:

1. For how many km altitude are AEF measurements made?

2. How was the electric field measured?

3. If the electric field is measured through the ionosphere, it must be freed from space weather effects for the accuracy of the results obtained. In fact, this is the point where I approach with the most skepticism in the study. Although the authors mention some information about the ionosphere in the conclusion part, it is seen that the effect of space weather is not taken into account in this study. Therefore, the reliability of the electric field data used in the study may be low. When the results of the authors are examined, the necessity of including space weather conditions in the study emerges.

Besides these basic problems, the quality of the shapes is not good. The axes of the figures are not readable.

What is the data sample range? such as 30 second, 5 minute intervals.

Author Response

Thank you so much for your kind comments and suggestions regarding the figures.

We have revised our manuscript in light of your suggestions.

1. For how many km altitude are AEF measurements made?

Thank you so much for your comment.

All our AEF measurements are made around the ground surface level. We carefully mounted the sensor on the ground in Kakioka (a flat field away from the tall conductive structure and data logger) so the observation height is ground level. The other two field mills are mounted on the top of the building (both are 8-story buildings) in Iwaki and Chofu (also away from the tall conductive structure and data logger). Therefore the heights of the observation are ~20 m from the ground.   

2. How was the electric field measured?

Thank you so much for your comment regarding our AEF measurements.

We use the Electric Field Mill (BOLTEK-100 system) to continuously record the vertical electric field with a sampling rate of 2 Hz (L88).

The principle of the system is shown below.

An electric field mill uses a mechanical chopper to alternately shield and expose several sense plates to an electric field. When the sense plates are exposed to the electric field an electric charge is drawn from the ground to the plates through a sense resistor. When the sense plates are shielded from the field the charge flows back to ground, again through the sense resistor. This moving charge is an electric current which is measured as an AC voltage across the sense resistor. The size of the voltage is proportional to the size of the electric field applied to the plates. 

Just in case, as is mentioned also in point 1, we carefully installed the sensor on the ground in Kakioka, on the top of the building in Iwaki and Chofu. We use an inverted downward-facing configuration to avoid precipitation noise and possible damage to the sensor unit (added in the manuscript L 89~90).

 3. If the electric field is measured through the ionosphere, it must be freed from space weather effects for the accuracy of the results obtained. In fact, this is the point where I approach with the most skepticism in the study. Although the authors mention some information about the ionosphere in the conclusion part, it is seen that the effect of space weather is not taken into account in this study. Therefore, the reliability of the electric field data used in the study may be low. When the results of the authors are examined, the necessity of including space weather conditions in the study emerges.

Thank you so much for your useful comment regarding space weather effect on AEF.

Indeed this is a very important point to consider to separate the known effect of AEF other than tropospheric weather.

We think that effects from the space weather seem to be small for our analyzed EQ events. The following space weather information has been added to the manuscript. Moreover, the subsection title for two EQs has been improved by adding the space weather condition (Subsections 3.2 and 3.4)

Around the date of both EQs space weather condition was rather quiet and so little effect will be expected for AEF from the space because Dst < 50 nT, Kp < 4, and no major high energy particle event (e.g. SPE (Solar Proton Event))  was observed. By no means, we will pay more attention to the space weather effect for future analysis. Thank you again.

4. Besides these basic problems, the quality of the shapes is not good. The axes of the figures are not readable.

Thank you so much for your useful comment regarding the figures.

We have improved the resolution of the figures.

5. What is the data sample range? such as 30 second, 5 minute intervals.

Thank you so much for your useful comment, we have added the following information regarding data analysis in the manuscript (L90~92).

At all stations, the electric field waveforms were observed with a time interval of  0.5-sec (2 Hz), but the average value per minute was used for the data analysis. 

Round 2

Reviewer 3 Report

The authors have made the necessary corrections considering the issues I have already mentioned. Although space weather conditions for the EQ1 are not exactly quiet, this assumption will not affect their results much.